behaviour/evolution/mathematical modelling

SARS-CoV-19, economic burden, shield immunity, evolutionary game theory

**Author for correspondence:**
K. M. Ariful Kabir
e-mail: k.ariful@yahoo.com

# Evolutionary game theory modelling to represent the behavioural dynamics of economic shutdowns and shield immunity in the COVID-19 pandemic

K. M. Ariful Kabir[1,3] and Jun Tanimoto[1,2]

[1]Interdisciplinary Graduate School of Engineering Sciences, and [2]Faculty of Engineering Sciences, Kyushu University, Kasuga-koen, Kasuga-shi, Fukuoka 816-8580, Japan
[3]Department of Mathematics, Bangladesh University of Engineering and Technology, Dhaka, Bangladesh

KMAK, 0000-0003-0249-5417

The unprecedented global spread of COVID-19 has prompted dramatic public-health measures like strict stay-at-home orders and economic shutdowns. Some governments have resisted such measures in the hope that naturally acquired shield immunity could slow the spread of the virus. In the absence of empirical data about the effectiveness of these measures, policymakers must turn to epidemiological modelling to evaluate options for responding to the pandemic. This paper combines compartmental epidemiological models with the concept of behavioural dynamics from evolutionary game theory (EGT). This innovation allows us to model how compliance with an economic lockdown might wane over time, as individuals weigh the risk of infection against the certainty of the economic cost of staying at home. Governments can, however, increase spending on social programmes to mitigate the cost of a shutdown. Numerical analysis of our model suggests that emergency-relief funds spent at the individual level are effective in reducing the duration and overall economic cost of a pandemic. We also find that shield immunity takes hold in a population most easily when a lockdown is enacted with relatively low costs to the individual. Our qualitative analysis of a complex model provides evidence that the effects of shield immunity and economic shutdowns are complementary, such that governments should pursue them in tandem.

# 1. Introduction

The COVID-19 pandemic presents urgent policy questions that must be addressed with modelling, as no vaccine is available and data about the pandemic is scarce [1–4]. Policy actions like quarantines, mobility restrictions, self-isolation and social distancing have been attempted across the globe, and empirical evaluations of their effectiveness are difficult with such an unprecedented virus [5–9]. The effectiveness of such measures will be affected by the socio-economic cost of those measures, which can be modelled in game-theoretic terms. As the pandemic progresses, shield immunity may begin to emerge if recovered patients are immune to infection or if a vaccine is developed. Policy decisions about vaccination and relaxing of stricter social measures can be informed by game theory models [10]. In hopes of modelling the social-learning aspect [11,12] of decision-making within the epidemic more realistically, we have followed Bauch in modelling decisions to vaccinate or self-isolate according to the imitation dynamic [13], a concept drawn from evolutionary game theory (EGT) [14,15].

The COVID-19 pandemic has caused more than 8 000 000 confirmed cases and more than 440 000 deaths across the globe as of 16 June 2020 [16]. Among public-health strategies aimed at suppressing and controlling the spread of the virus [17], strict stay-at-home orders have been essential policy tools [18]. Such a strict order has severe economic costs, as businesses close, services are reduced, unemployment increases and individuals' incomes are slowed [19]. If strict shutdowns are ordered without social support to mitigate the costs incurred by individuals, an epidemic may continue to spread among the disadvantaged despite shutdown measures [19]. Japan [20], the United States and the United Kingdom [21], for example, have implemented several emergency-relief packages that benefit people and businesses affected by the COVID-19 pandemic. These measures highlight economic disparities among countries, and those with fewer resources are less able to provide social support. In a society with limited access to social supports, individuals' compliance with stay-at-home orders requires them to balance the risk of financial damage while staying at home against the risk of infection if deciding not to stay at home. The outcomes of these individual-level decisions directly affect the society-level effectiveness of stay-at-home orders. So, the relationship between the economic costs of a social lockdown and individuals' likelihood to comply with that lockdown is a relevant relationship to model to better understand policy responses to the COVID-19 pandemic.

EGT [22–27] provides a framework for explaining individual behaviours in a social setting in which individual preferences depend on a variety of risks. This paper proposes a relatively complex game-theoretic model that is intended to give a realistic picture of how the cost of policy actions determines their effectiveness. The model gives insight into how the spread of an epidemic is affected by both the individual economic costs of public-health measures and the real risk of infection. The effects of a lockdown's costs are modelled with a counter-compliance factor [28], which reflects how individuals' fatigue with a long-term lockdown might increase their tendency to go out, thereby increasing the spread of the infection. The risk of infection is modelled with a term for shield immunity [29], which reduces the spread of infection as more recovered or vaccinated individuals are no longer at risk of spreading the disease.

# 2. Model and methods

## 2.1. Behavioural model

Our behavioural model assumes that players in a game must choose whether to comply with a stay-at-home order intended to stop the spread of an epidemic. The stay-at-home order has some economic cost. The players of the game are individuals exposed to the virus, who adopt the strategy of compliance or non-compliance (infection) with the public-health measure. When choosing a strategy, the pay-off depends on a balance of the perceived pay-off of compliance $[-C_Q \cdot Q(t)]$ against the pay-off of risking infection through non-compliance $[-C_i \cdot I^{\text{tot}}(t)]$. $C_Q$ is the economic cost of stay-at-home, $C_i$ is the cost of infection, $Q(t)$ is the perceived fraction of quarantined and non-infected individuals over time $t$ and $I^{\text{tot}}(t)$ is the total number of infected individuals ($I^{\text{tot}}(t) = I^S(t) + I^A(t)$, which is the sum of symptomatic $I^S$ and asymptomatic $I^A$ infected at time $t$). The expected pay-off for changing strategies can be measured as $-C_Q \cdot Q(t) + C_i \cdot I^{\text{tot}}(t)$. This term appears in the derivative of the time evolution of the rate at which individuals choose to risk infection, $\eta$, as follows:

$$\dot{\eta} = m \cdot \eta(t) \cdot [1 - \eta(t)][-C_Q \cdot Q(t) + C_i \cdot I^{\text{tot}}(t)]. \qquad (2.1)$$

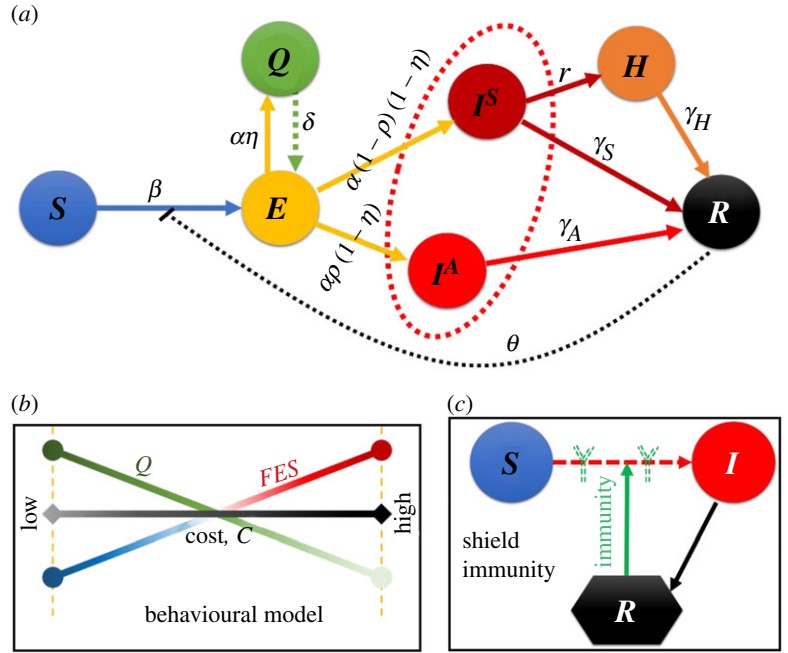

**Figure 1.** Schematic of behavioural epidemiological model with shield immunity. (*a*) epidemic dynamics of susceptible–exposed–quarantined–infected (asymptomatic & symptomatic)–hospitalized–recovered, (*b*) individual-based behavioural model and (*c*) scenario of shield immunity.

In this equation, $m$ is a proportionality constant that converts a portion of individuals into the transmission rate. $\eta(t)$ is the rate at which individuals choose to take quarantine to reduce the risk of infection at time $t$. If we assume for simplicity's sake that the relative economic costs and infection are proportional, i.e. $C = C_Q/C_i$, and we set $C_i = 1$ and $0 \leq C \leq 1$, then equation (2.1) simplifies to

$$\dot{\eta} = m \cdot \eta(t) \cdot [1 - \eta(t)] \cdot [-C \cdot Q(t) + I^{\mathrm{tot}}(t)]. \tag{2.2}$$

## 2.2. Epidemic dynamics

Previous uses of social-learning behavioural models to study epidemics have focused on vaccination and treatment games, often using mean-field approximations [24–27]. We use the mean-field approximation technique to solve our model's system of equations, as it is stochastic in nature. We build upon the well-known susceptible–exposed–infected–recovered (SEIR) compartmental model of epidemic dynamics to calculate the numbers of quarantined and infected individuals ($Q(t)$ and $I^{\mathrm{tot}}(t)$). We add terms for quarantined and hospitalized individuals to produce a SEQIHR (susceptible–exposed–quarantined–infected–hospitalized–recovered) compartmental model. The model is expressed by the following set of differential equations, whose relationships are diagrammed in figure 1:

$$\dot{S} = -\frac{\beta S(I^A + I^S + q(C, t)Q + hH)}{1 + \theta R}, \tag{2.3}$$

$$\dot{E} = \frac{\beta S(I^A + I^S + q(C, t)Q + hH)}{1 + \theta R} - \alpha E + \delta Q, \tag{2.4}$$

$$\dot{Q} = \alpha \eta(t)E - \delta Q, \tag{2.5}$$

$$\dot{I}^S = \alpha(1 - \rho)(1 - \eta(t))E - rI - \gamma_s I^S, \tag{2.6}$$

$$\dot{I}^A = \alpha \rho(1 - \eta(t))E - \gamma_A I^A, \tag{2.7}$$

$$\dot{H} = rI - \gamma_h H, \tag{2.8}$$

$$\dot{R} = \gamma_s I^S + \gamma_A I^A + \gamma_h H. \tag{2.9}$$

Here, $S$, $E$, $Q$, $I^S$, $I^A$, $H$ and $R$ are the fractions of the population that are susceptible, exposed, quarantine, symptomatic infected, asymptomatic infected, hospitalized and recovered, respectively. $\beta$ is the infection's transmission rate, $\alpha$ is the rate at which individuals progress from exposed to quarantined or exposed to infected ($E$ to $Q$ or $I^*$), $\delta$ is the rate at which individuals change

**Table 1.** Default parameter values and varied parameters.

| parameter | description | values | references |
|---|---|---|---|
| $C$ | relative cost of lockdown | [0,1] | (varied) |
| $h$ | hospital facilities factor | 1.0 | (varied) |
| $q$ | public counter-compliancy factor | 1.0 | (varied) |
| $r$ | testing rate/hospitalized rate | 0.1 day$^{-1}$ | (varied) |
| $\alpha$ | incubation rate to be infective | 1/6 day$^{-1}$ | [7,28] |
| $\beta$ | transmissibility rate | 2.0 person day$^{-1}$ | [7,28] |
| $\gamma_a$ | recovery rate (from asymptomatic) | 1/6 day$^{-1}$ | [28] |
| $\gamma_h$ | recovery rate (from hospital) | 1/18 day$^{-1}$ | [30] |
| $\gamma_s$ | recovery rate (from symptomatic) | 1/10 day$^{-1}$ | [28] |
| $\delta$ | quarantine to exposed rate | 1/30 day$^{-1}$ | assumed |
| $\eta$ | self-quarantine rate | 0.1 day$^{-1}$ | (varied) |
| $\theta$ | shield-immunity factor | 0 | (varied) |
| $\rho$ | asymptomatic infection rate | 0.5 day$^{-1}$ | assumed |

compartments from quarantined to exposed ($Q$ to $E$), $r$ is the rate at which individuals change compartments from symptomatic infected to hospitalized ($I^s$ to $H$), and $\gamma_s$, $\gamma_A$ and $\gamma_h$ are the recovery rates for symptomatic infected, asymptomatic infected and hospitalized individuals, respectively. The tendency to not comply is modelled with $q$ and the tendency that an infected individual is hospitalized is $h$. Shield immunity is modelled with the term $(1 + \theta R)$; when $\theta = 0$, shield immunity has no effect.

## 2.3. Counter-compliance

Our previous work on this model assumed that the counter-compliance factor $q$ [28] is constant over time. This factor is likely to change over time, however—if a lockdown stays in place for a very long time, people may become more inclined to defect. This likelihood will depend on the cost of the lockdown $C$. Appearing in equations (2.3) and (2.4), we formulate $q$ as a function of cost and time as follows:

$$q(C, t) = C \cdot [1 - \exp(-q_0 t)]. \tag{2.10}$$

We test three different versions of this model to qualitatively compare the model behaviour with different factors included. In **phase 0**, each of the parameters are taken from table 1, and the model behaviour serves as a baseline. **Phase 1** tests the behavioural model in equation (2.2) in isolation, with $q = 1.0$. **Phase 2** varies both $\eta$ and $q$ as given by equations (2.2) and (2.10), respectively.

To numerically solve the above stated set of equations, fourth-order Runge–Kutta method is used. Initially, we presumed the initial values as, $S(0) \approx 1.0$, $E(0) = 0$, $Q(0) = 0$, $I^S(0) \approx 0$, $I^A(0) \approx 0$, $H(0) = 0$ and $R(0) = 0$.

# 3. Result and discussion

This section presents the outcomes of numerical simulations of our model, which combines the classic compartmental epidemic model with behaviour dynamics from EGT. Figure 2$a$ plots the fraction of total infected individuals over time. Figure 2$b$ plots the final epidemic size (FES) at equilibrium, $R(\infty)$, against the cost of lockdown $C$. These plots confirm that all models reach a stable equilibrium even though the social-learning dynamic generates different behaviour for different values of $C$ (0.0, 0.5, 1.0) (figure 2$a$; phase 2).

When the model includes some effect of the cost of lockdown, the fraction of total infected individuals is less than it is in the default case (phase 0; black). As figure 2$a$ shows, increasing the cost causes the infection to spread further, as expected. Also as expected, the lockdown measure with the lowest economic costs is most effective. We note that the individual-level economic cost of a lockdown can be mitigated with social-relief policies like unemployment insurance and direct cash disbursements.

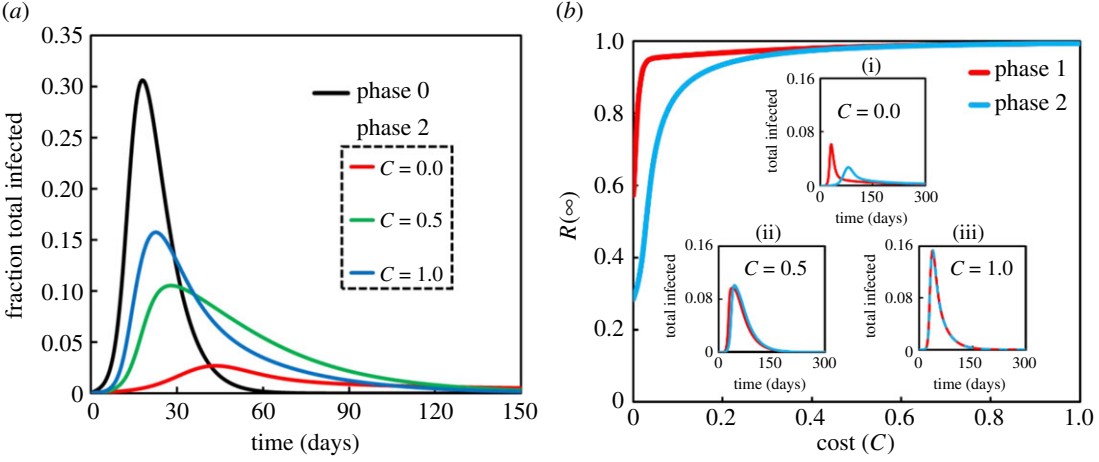

**Figure 2.** (a) Comparison of behavioural-dynamics model with default case. The parameters common to both models are listed in table 1. The cost of COVID-19 lockdown is $C = 0.0$ (red), $C = 0.5$ (green), and $C = 1.0$ (blue). (b) Final epidemic size $R(\infty)$ against the cost of lockdown, $C$ for phase 1 (red) and phase 2 (blue) models. Insets show sensitivity analysis of total infected individuals to show how different costs (i) $C = 0.0$, (ii) $C = 0.5$ and (iii) $C = 1.0$ affect difference in the behaviours of the phase 1 and 2 models.

The basic results above confirm that our term for counter-compliance affects the epidemic dynamics in our model (figure 2a). Now we turn to how the equilibrium extent of infection (FES, $R(\infty)$) behaves as a function of C with the different behavioural dynamics implemented ($\eta$ and $q$) (figure 2b).

When the cost C is low, the models show some interesting qualitative differences. In phase 1, individuals' tendency toward counter-compliance is constant at $q = 1.0$, meaning that individuals have no incentive to self-isolate; in this case, the FES is highest. However, in phase 2 in which $q$ depends on both cost and time, FES is lower in the phase 2 model for small C and the models behave the same for higher C.

These results show that our inclusion of behavioural dynamics yields a model that accords with realistic assumptions about human behaviour. We expect that people will adhere strictly to a stay-at-home measure at first, but will begin to relax their compliance over time, especially if staying at home is costly. At the same time, we expect that people will be more likely to comply with a stay-at-home order if its cost is low, perhaps because of social programmes enacted alongside the stay-at-home order. This qualitative analysis of our model shows that complex behavioural-dynamics models can be useful for assessing the effectiveness of public-health policy in a pandemic.

To observe how several parameter settings influence the incidence of the peak fraction of hospitalized individuals as well as the FES along with cost at equilibrium, we portrayed figure 3. In particular, we present four panels that vary different parameters: (a) $m$ (0.0, 0.1, 0.5, 1.0), (b) $r$ (0.1, 0.5, 1.0), (c) $h$ (0.1, 0.5, 1.0) and (d) $\rho$ (0.1, 0.5, 1.0). We interpret these plots as follows:

(i) (a): if individuals imitate strategies more readily (larger $m$), the epidemic dynamics are more sensitive to C; the lower cost reduces the FES, which in turn lessens the peak of hospitalized individuals. However, if $m = 0$, FES and peak-hospitalized are unaffected by C.

(ii) (b): increasing the testing rate $r$ increases the number of hospitalized, though the peak of hospitalizations remains unchanged.

(iii) (c): the FES decreases as access to hospitals $h$ improves. Yet, it can be observed that the fraction of peak-hospitalized is less affected by increasing $h$.

(iv) (d): changing the asymptomatic rate $\rho$ from low to high lessens the outcome of peak-hospitalized individuals; however, the FES decreases slightly.

Next, we investigate the model with shield immunity included. The peak hospitalizations and fractions of infected and susceptible at equilibrium are plotted against the shield immunity rate $\theta$ for different costs in figure 4: (a) $C = 1.0$, (b) $C = 0.5$ and (c) $C = 0.0$. Figure 4a suggests that sufficiently small $\theta$ (less shield immunity) leads to the maximum values for FES $R(\infty)$ and peak-hospitalized $H$(max). Higher $\theta$ tends to reduce FES and increase the amount of free riders $S(\infty)$ that reduces hospital burden. Figure 4c represents the situation in which staying at home has no cost, and we see that individuals stay at home if shield immunity is high enough. In this case, free riding and breaking the stay-at-home order has no benefit.

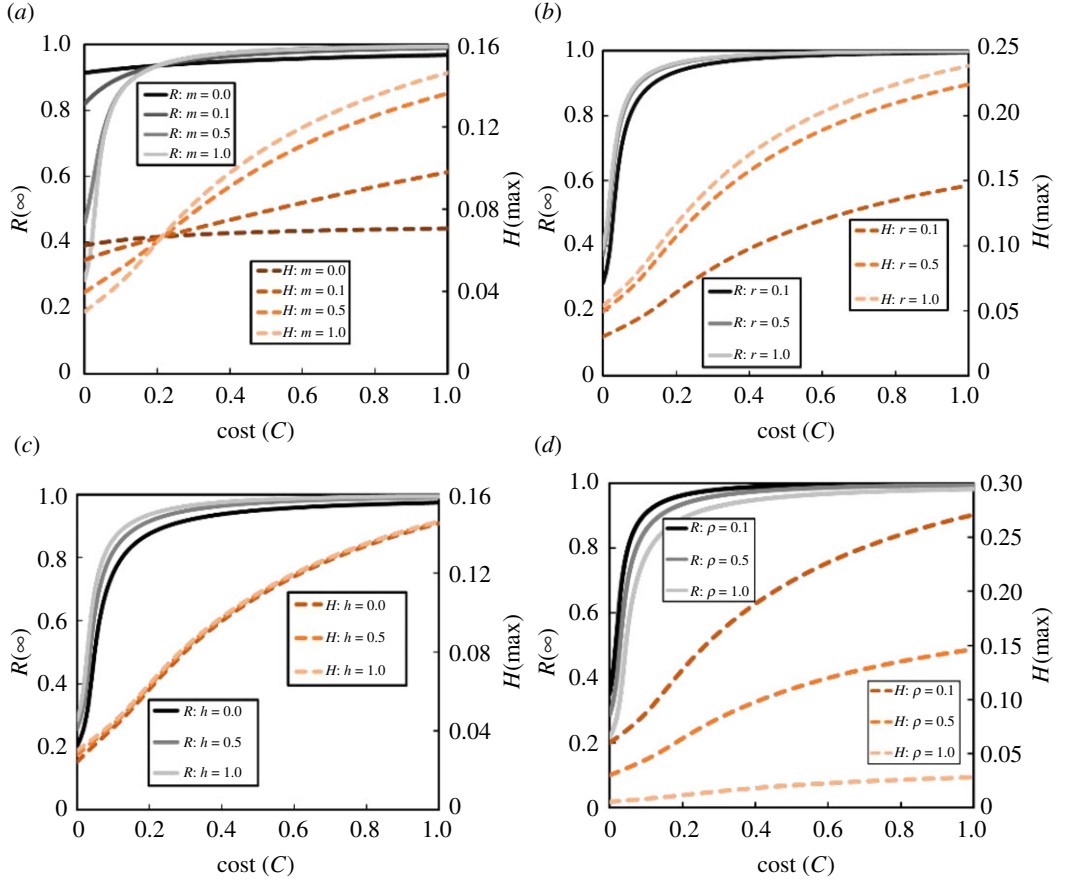

**Figure 3.** Impact of cost $C$ on the final epidemic size ($R(\infty)$) and the peak fraction of hospitalized ($H(\mathrm{max})$). (a) Proportionality constant $m = 0.0, 0.1, 0.5, 1.0$, (b) rate of testing $r = 0.1, 0.5, 1.0$, (c) rate of hospital facilities $h = 0.1, 0.5, 1.0$ and (d) rate of asymptomatic infection $\rho = 0.1, 0.5, 1.0$. All other parameters are as listed in table 1.

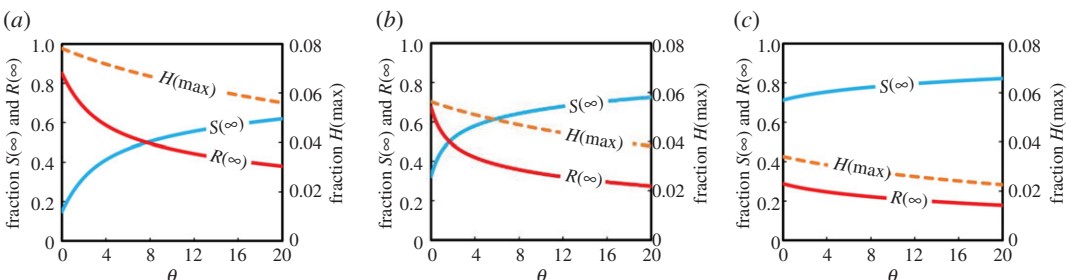

**Figure 4.** Impact of shield immunity factor, $\theta$ on the final epidemic size ($R(\infty)$) coloured red, suspected susceptible ($S(\infty)$) coloured blue and the peak fraction of hospitalized ($H(\mathrm{max})$) coloured orange. (a) $C = 1.0$, (b) $C = 0.5$ and (c) $C = 0.0$. Other parameter settings are listed in table 1.

As a final step, figure 5 presents a two-dimensional phase–plane analysis that varies the cost $C$ and transmission rate $\beta$ for a range of shield immunity factors $\theta$; (i) $\theta = 0$, (ii) $\theta = 5$, (iii) $\theta = 10$ and (iv) $\theta = 20$. In all cases, shield immunity affects the FES and peak hospitalizations significantly. As the values of both $C$ and $\beta$ increase, the FES as well as $H(\mathrm{max})$ grows remarkably as expected, since individuals are less likely to take quarantine when the cost is high, and individuals are more likely to be infected if the transmission rate is high. Meanwhile, natural shield immunity has a significant effect if and only if both the shield immunity factor is high and the cost of staying at home is low. This finding indicates that the effects of stay-at-home orders and shield immunity complement each other, but only if the cost of a stay-at-home order is sufficiently low and the tendency toward shield immunity is sufficiently high. Policymakers can affect the cost of staying at home, but the effectiveness of shield immunity is determined by the nature of the virus. Since no one yet knows if or for how long recovered

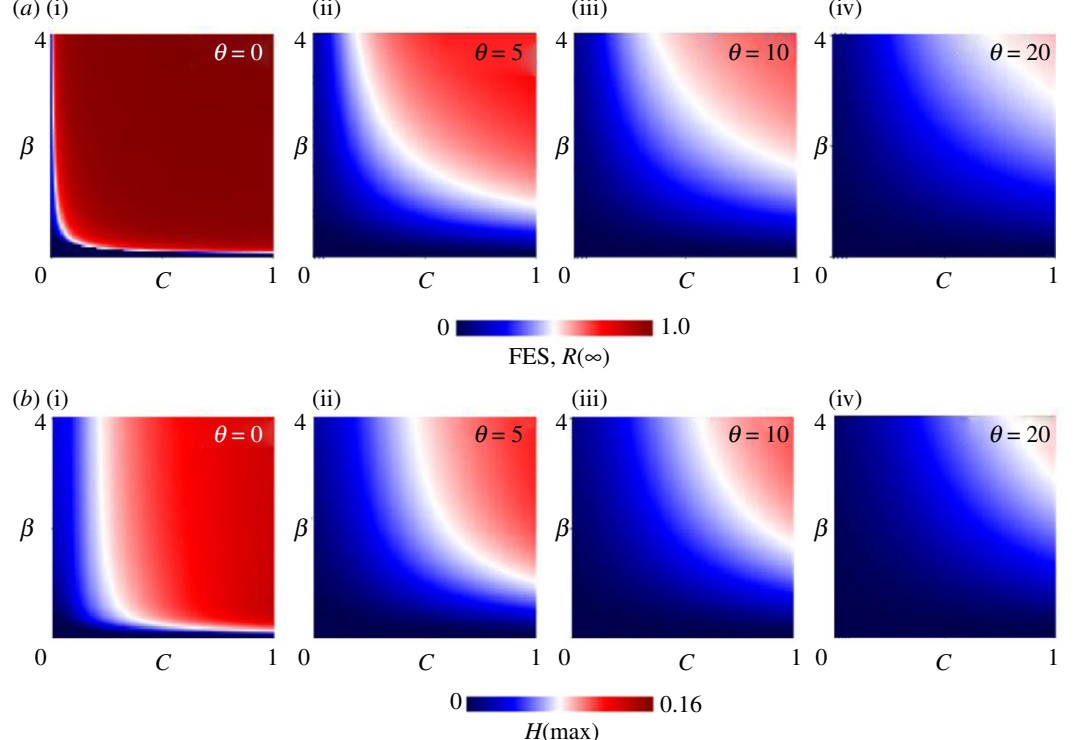

**Figure 5.** (*a*) Final epidemic size FES, $R(\infty)$ and (*b*) the peak fraction of hospitalized individuals $H$(max) along $C$ versus $\beta$.

individuals are immune to COVID-19, our models suggest that policymakers should do everything possible to lessen the cost of staying at home.

## 4. Conclusion

Our qualitative analyses of increasingly complex models suggest that complex social-learning dynamics can be captured in compartmental epidemic models that include game-theoretic concepts of imitation in an evolutionary game. Our parametric analysis suggests that naturally acquired shield immunity is unlikely to be effective in quelling an epidemic in the absence of social control measures that carry reasonably low costs for individuals. The results discussed above demonstrate the feasibility of analysing complex models of epidemics and social learning. With further development, we expect that such a model will prove helpful in developing insights about how to balance the economic costs of social controls as governments continue to navigate their responses to the COVID-19 pandemic.

Data accessibility. This article has no additional data.

Authors' contributions. K.M.A.K. conceived the presented idea and developed the theory and performed the computations. J.T. encouraged K.M.A.K. to investigate and supervised the findings of this work.

Competing interests. We declare we have no competing interests.

Funding. This study was partially supported by a Grant-in-Aid for Scientific Research from JSPS, Japan, KAKENHI (grant nos. JP 19KK0262 and JP 20H02314) awarded to J.T.

Acknowledgements. We would like to express our gratitude to them.

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
