## [Reviewer comments · Royal Society Open Science]

Review History

RSOS-201095.R0 (Original submission)

Review form: Reviewer 1 (Md. Shahidul Islam)

Is the manuscript scientifically sound in its present form?

Yes

Are the interpretations and conclusions justified by the results?

Yes

Is the language acceptable?

Yes

Do you have any ethical concerns with this paper?

No

Have you any concerns about statistical analyses in this paper?

No

Recommendation?

Accept with minor revision (please list in comments)

Comments to the Author(s)

1. What are the initial values of each compartment presumed in simulation (S, E, ...)? It seems was not mentioned.
2. Is it possible to write down the payoff matrix of the game?
3. What is the numerical method to calculate the differential equations and which programming is used?

Review form: Reviewer 2 (Yap Yee Jiun)**Is the manuscript scientifically sound in its present form?**

Yes

Are the interpretations and conclusions justified by the results?

Yes

Is the language acceptable?

Yes

Do you have any ethical concerns with this paper?

No

Have you any concerns about statistical analyses in this paper?

No

Recommendation?

Accept as is

Comments to the Author(s)

The paper is well written, with important results for policy makers.

Review form: Reviewer 3**Is the manuscript scientifically sound in its present form?**

No

Are the interpretations and conclusions justified by the results?

Yes

Is the language acceptable?

Yes

Do you have any ethical concerns with this paper?

No

Have you any concerns about statistical analyses in this paper?

No

Recommendation?

Major revision is needed (please make suggestions in comments)

Comments to the Author(s)

This paper explores the evolutionary game theory of population adherence to lockdown in the face of ongoing COVID-19 transmission, by combining a model of human decision-making with a model of infection spread. The dynamics of decision-making in a population is an important but neglected aspect of the pandemic modelling literature, since the success of infection control during COVID-19 depends strongly upon the population's acceptance of measures like social distancing and lockdown. Hence, the work is very novel. But I have some concerns about the structure of the epidemic model. Below I make some recommendations for major revisions, as well as a few minor revisions. Besides these issues, the work appears methodologically sound and is well written.

Major Comments:

- The model assumes that individuals who have been infected (the unfortunately-named "exposed" category in epidemiological modelling literature) can recover immediately to the susceptible category upon clearing quarantine, but that is not possible since individuals who are infected must go through the entire infection process and end in the Recovered compartment with naturally-derived immunity. The model should make individuals progress from exposed to infectious quarantined, possibly also with hospitalization, and from there, to recovered.
- There is no need to encode herd immunity explicitly through the theta term in the denominator. Herd immunity is an outcome of compartmental models that arises naturally as the susceptible pool is gradually depleted through infection, and the proportion of recovered individuals increases. It should not be treated as an input parameter. Hence, the $1 + \theta \cdot R$ term in the denominator should be removed. It's possible that the authors had something closer to standard incidence in mind, but in that case, the functional form of the incidence term would be different.
- The model equations in the main text are differential equations but the appendix indicates a discrete-time model is being used. The authors should use a robust and well-validated numerical method to integrate the equations, such as 4th/5th order Runge-Kutta, instead of using a discrete-time model.
- Some of the durations of infectiousness seem to be too long. For instance, even though individuals may spend up to 30 days in the hospital, most of the infectiousness is concentrated in the first week and so assuming a duration of 30 days is an overestimate. The same is true for the other recovery rates. The authors could review the literature on the incubation period and serial interval of COVID-19 to find more realistic values.

Minor Comments:

- Page 6 line 6: conclusions section mentions a vaccination game, which may be a typo.
- Table 1: please provide the units of these parameters.
- The authors should also explain where their baseline parameter values come from.

Decision letter (RSOS-201095.R0)

Dear Mr KABIR,

The editors assigned to your paper ("Evolutionary game theory modeling to represent the behavioral dynamics of economic shutdowns and herd immunity in the COVID-19 pandemic") have now received comments from reviewers. We would like you to revise your paper in

accordance with the referee and Associate Editor suggestions which can be found below (not including confidential reports to the Editor). Please note this decision does not guarantee eventual acceptance.

Please submit a copy of your revised paper before 28-Aug-2020. Please note that the revision deadline will expire at 00.00am on this date. If we do not hear from you within this time then it will be assumed that the paper has been withdrawn. In exceptional circumstances, extensions may be possible if agreed with the Editorial Office in advance. We do not allow multiple rounds of revision so we urge you to make every effort to fully address all of the comments at this stage. If deemed necessary by the Editors, your manuscript will be sent back to one or more of the original reviewers for assessment. If the original reviewers are not available, we may invite new reviewers.

- Data accessibility

<http://datadryad.org/submit?journalID=RSOS&manu=RSOS-201095>

- Competing interests

- Authors' contributions

All submissions, other than those with a single author, must include an Authors' Contributions section which individually lists the specific contribution of each author. The list of Authors

should meet all of the following criteria; 1) substantial contributions to conception and design, or acquisition of data, or analysis and interpretation of data; 2) drafting the article or revising it critically for important intellectual content; and 3) final approval of the version to be published.

- Acknowledgements

- Funding statement

on behalf of Professor Matjaz Perc (Associate Editor) and Kevin Padian (Subject Editor)
openscience@royalsociety.org

Associate Editor's comments (Professor Matjaz Perc):

Many recent studies have been devoted to the COVID-19 pandemic, both in terms of forecasting as well as modelling. The current introduction does not do a very good job in giving a comprehensive review of this field of research. The authors should do better in this regard. See for example Early spread of COVID-19 in Romania: Imported cases from Italy and human-to-human transmission networks, *R. Soc. Open Sci.* 7, 200780 (2020) and Forecasting COVID-19, *Front. Phys.* 8, 127 (2020). Much research has also been devoted to the study of epidemics and vaccination in terms of evolutionary game theory, as reviewed for example in Statistical physics of vaccination, *Phys. Rep.* 664, 1-113 (2016).

Comments to Author:

Reviewers' Comments to Author:

Reviewer: 1

Comments to the Author(s)

1. What are the initial values of each compartment presumed in simulation (S, E,...)? It seems was not mentioned.
2. Is it possible to write down the payoff matrix of the game?

3. What is the numerical method to calculate the differential equations and which programming is used?

Reviewer: 2

Comments to the Author(s)

The paper is well written, with important results for policy makers.

Reviewer: 3

Comments to the Author(s)

This paper explores the evolutionary game theory of population adherence to lockdown in the face of ongoing COVID-19 transmission, by combining a model of human decision-making with a model of infection spread. The dynamics of decision-making in a population is an important but neglected aspect of the pandemic modelling literature, since the success of infection control during COVID-19 depends strongly upon the population's acceptance of measures like social distancing and lockdown. Hence, the work is very novel. But I have some concerns about the structure of the epidemic model. Below I make some recommendations for major revisions, as well as a few minor revisions. Besides these issues, the work appears methodologically sound and is well written.

Major Comments:

- The model assumes that individuals who have been infected (the unfortunately-named "exposed" category in epidemiological modelling literature) can recover immediately to the susceptible category upon clearing quarantine, but that is not possible since individuals who are infected must go through the entire infection process and end in the Recovered compartment with naturally-derived immunity. The model should make individuals progress from exposed to infectious quarantined, possibly also with hospitalization, and from there, to recovered.
- There is no need to encode herd immunity explicitly through the theta term in the denominator. Herd immunity is an outcome of compartmental models that arises naturally as the susceptible pool is gradually depleted through infection, and the proportion of recovered individuals increases. It should not be treated as an input parameter. Hence, the $1 + \theta \cdot R$ term in the denominator should be removed. It's possible that the authors had something closer to standard incidence in mind, but in that case, the functional form of the incidence term would be different.
- The model equations in the main text are differential equations but the appendix indicates a discrete-time model is being used. The authors should use a robust and well-validated numerical method to integrate the equations, such as 4th/5th order Runge-Kutta, instead of using a discrete-time model.
- Some of the durations of infectiousness seem to be too long. For instance, even though individuals may spend up to 30 days in the hospital, most of the infectiousness is concentrated in the first week and so assuming a duration of 30 days is an overestimate. The same is true for the other recovery rates. The authors could review the literature on the incubation period and serial interval of COVID-19 to find more realistic values.

Minor Comments:

- Page 6 line 6: conclusions section mentions a vaccination game, which may be a typo.
- Table 1: please provide the units of these parameters.
- The authors should also explain where their baseline parameter values come from.

Author's Response to Decision Letter for (RSOS-201095.R0)

See Appendix A.

RSOS-201095.R1 (Revision)

Review form: Reviewer 1 (Md. Shahidul Islam)

Is the manuscript scientifically sound in its present form?

Yes

Are the interpretations and conclusions justified by the results?

Yes

Is the language acceptable?

Yes

Do you have any ethical concerns with this paper?

No

Have you any concerns about statistical analyses in this paper?

No

Recommendation?

Accept as is

Comments to the Author(s)

In this dangerous situation of the pandemic, when everyone is fearing the current economic loss all around the world, as well as the possibility of a more devastating loss in the near future, this model is very relevant.

Mathematical reasoning is also impressive. The numerical simulation depicts the mathematical ideas proposed in the model nicely, which is praiseworthy.

Review form: Reviewer 3

Is the manuscript scientifically sound in its present form?

Yes

Are the interpretations and conclusions justified by the results?

Yes

Is the language acceptable?

Yes

Do you have any ethical concerns with this paper?

No

Have you any concerns about statistical analyses in this paper?

No

Recommendation?

Accept as is

Comments to the Author(s)

The authors have responded to my previous comments in a satisfactory way. This will make a nice contribution to the literature.

Decision letter (RSOS-201095.R1)

Dear Mr KABIR,

It is a pleasure to accept your manuscript entitled "Evolutionary game theory modeling to represent the behavioral dynamics of economic shutdowns and shield immunity in the COVID-19 pandemic" in its current form for publication in Royal Society Open Science. The comments of the reviewer(s) who reviewed your manuscript are included at the foot of this letter.

COVID-19 rapid publication process:

We are taking steps to expedite the publication of research relevant to the pandemic. If you wish, you can opt to have your paper published as soon as it is ready, rather than waiting for it to be published the scheduled Wednesday.

This means your paper will not be included in the weekly media round-up which the Society sends to journalists ahead of publication. However, it will still appear in the COVID-19 Publishing Collection which journalists will be directed to each week (<https://royalsocietypublishing.org/topic/special-collections/novel-coronavirus-outbreak>).

If you wish to have your paper considered for immediate publication, or to discuss further, please notify openscience_proofs@royalsociety.org and press@royalsociety.org when you respond to this email.

Please ensure that you send to the editorial office individual files for each figure and table included in your manuscript. You can send these in a zip folder if more convenient. Failure to provide these files may delay the processing of your proof. You may disregard this request if you have already provided these files to the editorial office.

on behalf of Professor Matjaz Perc (Associate Editor) and Kevin Padian (Subject Editor)
openscience@royalsociety.org

Reviewer comments to Author:
Reviewer: 3

Comments to the Author(s)
The authors have responded to my previous comments in a satisfactory way. This will make a nice contribution to the literature.

Reviewer: 1

Comments to the Author(s)
In this dangerous situation of the pandemic, when everyone is fearing the current economic loss all around the world, as well as the possibility of a more devastating loss in the near future, this model is very relevant.
Mathematical reasoning is also impressive. The numerical simulation depicts the mathematical ideas proposed in the model nicely, which is praiseworthy.

Appendix A

Response Letter to the Editor and the Reviewers

Editor's comment: We hope you are keeping well at this difficult and unusual time. We continue to value your support of the journal in these challenging circumstances. If Royal Society Open Science can assist you at all, please don't hesitate to let us know at the email address below.

The editors assigned to your paper ("Evolutionary game theory modeling to represent the behavioral dynamics of economic shutdowns and herd immunity in the COVID-19 pandemic") have now received comments from reviewers. We would like you to revise your paper in accordance with the referee and Associate Editor suggestions which can be found below (not including confidential reports to the Editor). Please note this decision does not guarantee eventual acceptance.

Reply: We considered all the editors and reviewers' comment with utmost seriousness and tried to adjust the manuscript accordingly where suitable. We also provided detailed response aimed at resolving the issues raised any concerns. We hope that the revised MS and this rebuttal response seem persuasive.

Report of the Associate Editor's

Comment: Many recent studies have been devoted to the COVID-19 pandemic, both in terms of forecasting as well as modelling. The current introduction does not do a very good job in giving a comprehensive review of this field of research. The authors should do better in this regard. See for example Early spread of COVID-19 in Romania: Imported cases from Italy and human-to-human transmission networks, R. Soc. Open Sci. 7, 200780 (2020) and Forecasting COVID-19, Front. Phys. 8, 127 (2020). Much research has also been devoted to the study of epidemics and vaccination in terms of evolutionary game theory, as reviewed for example in Statistical physics of vaccination, Phys. Rep. 664, 1-113 (2016).

Reply: We appreciate the editor's kind suggestion in terms of lack of our citation covering some previous works. Literatures suggested above have been added. Let us thank again.

3. Hâncean, M.G., Perc, M., Lerner, J.; Early spread of COVID-19 in Romania: imported cases from Italy and human-to-human transmission networks, R. Soc. Open Sci. 7, 200780 (2020).

4. Perc, M., Miksic', N. G., Slavinec, M., Stožer, A; Forecasting COVID-19, Front. Phys. 8, 127 (2020).

23. Z. Wang, CT. Bauch, S. Bhattacharyya, A. d'Onofrio, P. Manfredi, M. Perc, N. Perra, M. Salathé, 419 and D. Zhao, Statistical physics of vaccination. Phys. Rep. 664: 1-13 (2016).

Report of the First Referee

Comment:

1. What are the initial values of each compartment presumed in simulation (S, E,...)? It seems was not mentioned.
3. What is the numerical method to calculate the differential equations and which programming is used?

Reply: Let us really appreciate giving meaningful suggestion. With respect to the first and third point the reviewer questioning, we include some text in the model and methods section.

To numerically solve the above stated set of equations, 4th order Runge-Kutta method is used. Initially, we presumed the initial values as, $S(0) \approx 1.0$, $E(0) = 0$, $Q(0) = 0$, $I^S(0) \approx 0$, $I^A(0) \approx 0$, $H(0) = 0$ and $R(0) = 0$.

Comment:

2. Is it possible to write down the payoff matrix of the game?

Reply: We highly appreciate the reviewer meaningful comment. What the reviewer pointing out is somehow possible and our current formulation of imitation dynamic can be presented in a form of payoff matrix. Yet, what we took along our process, which somehow follows some pioneer studies like Bauch [11], was not going through establishing a payoff matrix, which interprets the game structure, in a sense, into a static formulation. Aside from the payoff matrix, what we considered was a social learning

process, that is not static but dynamic, based on the imitation dynamic of evolutionary game theory called “behavioral model”. The term $-C_Q \cdot Q(t) + C_i \cdot I^{tot}(t)$ is the payoff gain for switching strategies and its sign determines whether quarantine or infection is the favored switch. Suppose ΔE is the gain payoff between two strategies: E_Q , payoff to quarantine and E_i , payoff to infected. To evaluate these two strategies, we assign an expected payoff to each strategy as, $E_Q = -C_Q \cdot Q(t)$ and $E_i = -C_i \cdot I^{tot}(t)$. Hopefully these will enough to realize how the payoff matrix express into imitation dynamics used in the evolutionary game aspects.

Thank you for your understanding.

Report of the second Referee

Comment: The paper is well written, with important results for policy makers.

Reply: We are grateful for the positive tone of this comment. Hopefully, this will be enough to deem the manuscript acceptable for publishing.

Report of the third Referee

Comments: This paper explores the evolutionary game theory of population adherence to lockdown in the face of ongoing COVID-19 transmission, by combining a model of human decision-making with a model of infection spread. The dynamics of decision-making in a population is an important but neglected aspect of the pandemic modelling literature, since the success of infection control during COVID-19 depends strongly upon the population’s acceptance of measures like social distancing and lockdown. Hence, the work is very novel. But I have some concerns about the structure of the epidemic model. Below I make some recommendations for major revisions, as well as a few minor revisions. Besides these issues, the work appears methodologically sound and is well written.

Reply: The reviewer fairly summarized our main results. We are grateful for the overall positive tone of the comment. This is a great motivation to keep up the good work.

Major Comments:

Comments: The model assumes that individuals who have been infected (the unfortunately-named “exposed” category in epidemiological modelling literature) can recover immediately to the susceptible category upon clearing quarantine, but that is not possible since individuals who are infected must go through the entire infection process and end in the Recovered compartment with naturally-derived immunity. The model should make individuals progress from exposed to infectious quarantined, possibly also with hospitalization, and from there, to recovered.

Reply: We highly appreciate the meaningful suggestions. The reviewer’s pointing out about the compartment E (exposed) in which individuals are “infected” but not “infectious” yet. We apologize for our unpleasant model structure, which has been fixed. According to the reviewer’s suggestion, we modified our model structure and results. I think the current form of model appropriately depicted the COVID 19 based epidemiological aspect. Thus, we modestly request the reviewer’s patience and understanding. Let alone, we have thoroughly re-obtained all numerical results according to the modification inspired by the reviewer’s suggestion. Let us appreciate the reviewer’s important suggestion. To explain this, we modified the model in the revised manuscript:

The model is expressed by the following set of differential equations, whose relationships are diagrammed in figure 1.

$$\dot{S} = -\frac{\beta S(I^A + I^S + q(C, t)Q + hH)}{1 + \theta R}, \quad (2.1)$$

$$\dot{E} = \frac{\beta S(I^A + I^S + q(C, t)Q + hH)}{1 + \theta R} - \alpha E + \delta Q, \quad (2.2)$$

$$\dot{Q} = \alpha \eta(t) E - \delta Q, \quad (2.3)$$

$$\dot{I}^S = \alpha(1 - \rho)(1 - \eta(t))E - rI - \gamma_S I^S, \quad (2.4)$$

$$\dot{I}^A = \alpha \rho(1 - \eta(t))E - \gamma_A I^A, \quad (2.5)$$

$$\dot{H} = rI - \gamma_h H, \quad (2.6)$$

$$\dot{R} = \gamma_S I^S + \gamma_A I^A + \gamma_h H. \quad (2.7)$$

δ is the rate at which individuals change compartments from quarantined to exposed (Q to E),

Figure 1

Comments: There is no need to encode herd immunity explicitly through the theta term in the denominator. Herd immunity is an outcome of compartmental models that arises naturally as the susceptible pool is gradually depleted through infection, and the proportion of recovered individuals increases. It should not be treated as an input parameter. Hence, the $1 + \theta R$ term in the denominator should be removed. It's possible that the authors had something closer to standard incidence in mind, but in that case, the functional form of the incidence term would be different.

Reply: Upon rereading the manuscript in light of the reviewer's comment, it became apparent to us that the term we used “herd immunity” is misleading us to how the naturally immunized people affect to control the disease. According to previous well-established work by Weitz et. al [30], we replace “shield immunity” instead of “herd immunity”. Because the described concept is new, we therefore ask the reviewer for a little patience. Thank you for your understanding.

Comments: The model equations in the main text are differential equations but the appendix indicates a discrete-time model is being used. The authors should use a robust and well-validated numerical method to integrate the equations, such as 4th/5th order Runge-Kutta, instead of using a discrete-time model.

Reply: What a delightful idea. According to reviewer's suggestion, we recalculated all the numerical simulation by using 4th order Runge-Kutta method instead of discrete model. But unfortunately, we didn't find any significant difference between our current results (discrete) and 4th order RK results. Thank you for your understanding.

Note: In the revised version, we attached the 4th order RK code.

Comments: Some of the durations of infectiousness seem to be too long. For instance, even though individuals may spend up to 30 days in the hospital, most of the infectiousness is concentrated in the first week and so assuming a duration of 30 days is an overestimate. The same is true for the other recovery rates. The authors could review the literature on the incubation period and serial interval of COVID-19 to find more realistic values.

Reply: We appreciate the reviewer’s sentiment in terms of lack of appropriate citation covering some previous works that somehow mislead reviewer. We also sorry for some assumed values, which have been already fixed in Table 1. Our main intention is to model and analyze the general formula based on the human decision-making process (not data fitting). So, it’s tough to presume specific values for each parameter that can display realistic situations. For simplicity, we assumed the utmost condition for different transmission rates by following previous works. However, if the reviewer feels that we are leaving out some truly vital studies, please suggest them to us, and we will be more than happy to include them in the manuscript. We have already corrected Table 1.

Parameter	Description	Previous values	New values
γ_a	Recovery rate (from asymptomatic)	1/10 day ⁻¹	1/6 day ⁻¹
γ_h	Recovery rate (from hospital)	1/30 day ⁻¹	1/18 day ⁻¹
γ_s	Recovery rate (from symptomatic)	1/20 day ⁻¹	1/10 day ⁻¹

Minor Comments:

Comments: Page 6 line 6: conclusions section mentions a vaccination game, which may be a typo.

Reply: Corrected.

Our qualitative analyses of increasingly complex models suggest that complex social-learning dynamics can be captured in compartmental epidemic models that include game-theoretic concepts of imitation in an evolutionary game.

Comments: Table 1: please provide the units of these parameters.

Reply: Added all relevant units in Table 1.

Comments: The authors should also explain where their baseline parameter values come from.

Reply: Added corresponding source of assumed and used data in Table 1.

In response to these comments, we revised the Table 1 as follows:

Table 1. Default parameter values and varied parameters-

Parameter	Description	values	References
\mathcal{C}	Relative cost of lockdown	[0,1]	(varied)
h	Hospital facilities factor	1.0	(varied)
q	Public counter-compliance factor	1.0	(varied)
r	Testing rate/ hospitalized rate	0.1 day ⁻¹	(varied)
α	Incubation rate to be infective	1/6 day ⁻¹	[7, 28]
β	Transmissibility rate	2.0 person day ⁻¹	[7, 28]
γ_a	Recovery rate (from asymptomatic)	1/6 day ⁻¹	[28]
γ_h	Recovery rate (from hospital)	1/18 day ⁻¹	[30]
γ_s	Recovery rate (from symptomatic)	1/10 day ⁻¹	[28]
δ	Quarantine to exposed rate	1/30 day ⁻¹	assumed
η	Self-quarantine rate	0.1 day ⁻¹	(varied)
θ	Shield -immunity factor	0	(varied)
ρ	Asymptomatic infection rate	0.5 day ⁻¹	assumed